# Molecular Characterization of Aquaporins Genes from the Razor Clam *Sinonovacula constricta* and Their Potential Role in Salinity Tolerance

Wenbin Ruan [1,2], Yinghui Dong [2], Zhihua Lin [2] and Lin He [2,*]

1   School of Marine Sciences, Ningbo University, Ningbo 315211, China; 1701091032@nbu.edu.cn
2   Key Laboratory of Aquatic Germplasm Resources of Zhejiang, Zhejiang Wanli University, Ningbo 315100, China; dongyinghui@zwu.edu.cn (Y.D.); linzhihua@zwu.edu.cn (Z.L.)
*   Correspondence: helin@zwu.edu.cn; Tel.: +86-188-5823-4884

**Abstract:** Aquaporins (AQPs) play crucial roles in osmoregulation, but the knowledge about the functions of AQPs in *Sinonovacula constricta* is unclear. In this study, *Sc-AQP1*, *Sc-AQP8*, and *Sc-AQP11* were identified from *S. constricta*, and the three *Sc-AQP*s are highly conserved compared to the known *AQP*s. The qRT-PCR analysis revealed that the highest mRNA expressions of *Sc-AQP1*, *Sc-AQP8*, and *Sc-AQP11* were detected in the gill, digestive gland, and adductor muscle, respectively. In addition, the highest mRNA expression of *Sc-AQP1* and *Sc-AQP11* was detected in the D-shaped larvae stage, whereas that of *SC-AQP8* was observed in the umbo larvae stage. The mRNA expression of *Sc-AQP*1, *Sc-AQP8*, and *Sc-AQP11* significantly increased to 12.45-, 12.36-, and 27.44-folds post-exposure of low salinity (3.5 psu), while only *Sc-AQP1* and *Sc-AQP11* significantly increased post-exposure of high salinity (35 psu) ($p < 0.01$). The fluorescence in situ hybridization also showed that the salinity shift led to the boost of *Sc-AQP1*, *Sc-AQP8*, and *Sc-AQP11* mRNA expression in gill filament, digestive gland, and adductor muscle, respectively. Knockdown of the *Sc-AQP1* and *Sc-AQP8* led to the decreased osmotic pressure in the hemolymph. Overall, these findings would contribute to the comprehension of the osmoregulation pattern of AQPs in *S. constricta*.

**Keywords:** *Sinonovacula constricta*; aquaporin; osmoregulation; RNA interference; salinity

## 1. Introduction

Aquaporins (AQPs) are one-of-a-kind small integral membrane proteins with transmembrane channels that have high water permeability in specific biological membranes [1]. They are responsible for sustaining net water movement at a rate suitable for fulfilling cellular or transcellular functions in animals, plants, and prokaryotes [2–5]. AQPs, also known as major integral proteins (MIPs), exist on the surface of the cell membrane as a tetramer [6]. Each AQP's monomer contains water pores [6] and commonly includes six membrane-spanning α-helices packed to form part of a trapezoid-like structure [7,8], which is relative to the water transportation.

To date, numerous kinds of *AQP* genes, with different classifications in their biological function in transporting water and other small molecules, have been identified in vertebrates and invertebrates [6]. For instance, at least 13 definite *AQP* isoforms have been identified in vertebrates, whereby *AQP*s can be divided into four grades corresponding to classical *AQP*, aquaglyceroporins (AQGPs), S-aquaporin, and *AQP-8* type ones [9]. In addition, the invertebrate MIPs are located in classical *AQP* and *AQP8*-type grades based on the evolutionary framework [6]. Nevertheless, only a few studies have considered the aquatic animals as the models; they can also represent an excellent animal model to investigate the water transport activity of AQP in response to the changeable ambient salinity [10].

The razor clam (*Sinonovacula constricta*) (Lamarck, 1818), a marine shellfish species with high commercial value, has been widely cultured in muddy intertidal zones due to its fast growth and high nutritional value [11]. Recently, the industry of razor clam has greatly expanded, but its growth is being affected significantly by the dramatic changes in salinity at rearing ponds, thereby resulting in enormous economic losses [12]. Significantly, the processes of ion and water transport in the gills lead to the variation of osmotic pressure [13]. The current researches are mostly concerned with studying membrane-bound ion channels and exchangers and intercellular tight junctions, giving rise to an advanced molecular pathway involved in ion transport [14]. However, little research has been conducted to investigate the molecular pathways of water transport in shellfish. Thus, there are still uncertainties at the molecular level regarding water transport mechanisms, especially across different anatomical sections of the organs, because different species have various water transport mechanisms [14].

In this study, to investigate the potential role of *AQP*s of *S. constricta* in salinity tolerance, three new *AQP*s isoforms from *S. constricta*, denoted as *Sc-AQP1*, *Sc-AQP8*, and *Sc-AQP11*, were cloned and characterized. Then, their expression profiles in different tissues and developmental stages were examined. Furthermore, RNA interference (RNAi) technology and fluorescence in situ hybridization (FISH) were used to explore the potential functions of three *Sc-AQP*s on osmotic pressure regulation of the hemolymph under different salinities.

## 2. Materials and Methods

### 2.1. Animal Cultivation

Healthy juvenile razor clams (mixed sexes) with an average shell length of $4.0 \pm 0.5$ cm were collected from Yinzhou Danyan Aquaculture Field (salinity of 18 psu, pH 8.2, 19–25 °C) in Ningbo, Zhejiang Province, China, and then cultured in the genetic breeding research center of Zhejiang Wanli University. These clams came from the same family line and shared an identical genetic background. The laboratory conditions set for clams' cultivation were similar to the collection point in the field. Collected clams were acclimatized in 100 L aerated seawater with a salinity level of 18 psu and pH range of 8–8.5 at $24 \pm 1$ °C. Five hundred and fifteen clams were kept in the laboratory for two weeks prior to the following experiments, including tissue distribution analysis (5 clams), salinity challenge experiments (300 clams), and RNA interference (210 clams). During the acclimatization period, *Chaetoceros calcitrans* (Takano, 1968) and *Platymonas helgolandica* Kylin var. tsingtaoensis were used as a diet source for razor clam. The seawater was refreshed twice daily after feeding the microalgae for 4 h. Embryos and larvae were reared in hatching tanks in 18 psu salinity seawater, fed with *Isochrysis galbana* Parke 8701 (1949) and *C. meiilleri* Lemmerman. During the period of animal cultivation, the pH, temperature, and salinity of seawater were monitored by using HD40 (HACH, Loveland, CO, USA) and BEC-600 (BELL, Dalian, Liaoning, China), respectively.

### 2.2. Tissue and Spatiotemporal Expression Analysis

Five clams were dissected to investigate the distribution of *Sc-AQP1*, *Sc-AQP8*, and *Sc-AQP11* in different tissues, including the gill, intestine, kidney, digestive gland, siphon, mantle, foot, and adductor muscle. Additionally, embryos and larvae from ten different developmental stages (unfertilized mature eggs, fertilized eggs, 4 cells, blastula, gastrulae, trochophore, D-shaped larvae, umbo larvae, eyebot larvae, and juvenile clams) were collected and immediately frozen in liquid nitrogen. Two hundred samples were included in each developmental stage. The samples were then stored at $-80$ °C prior to total RNA extraction and following quantitative real-time (qRT-PCR) analysis.

### 2.3. Low- and High-Salinity Challenges

Based on the 50% lethal dose ($LD_{50}$) test of salinity, the levels of salinity lethal to 50% of razor clam were determined to be 3.5 and 35 psu in 168 h (data not shown). Then the salinity levels of 3.5 and 35 psu were set as the low-salinity and high-salinity pressure.

To investigate the temporal expression profile of *Sc-AQP1*, *Sc-AQP8*, and *Sc-AQP11* after low-salinity and high-salinity treatment, razor clams were randomly divided into three groups, with each group containing 300 clams: two experimental groups (Group 1 and Group 2) and one control group. For Group 1, razor clams were transferred from 18 to 3.5 psu for the acute low-salinity stress, while razor clams in Group 2 were transferred from 18 to 35 psu for the acute high-salinity stress. The razor clams in the control group were continuously cultured in filtrated natural seawater (collected from Xiangshan harbour, Ningbo, Zhejiang province) of 18 psu. The 3.5 psu seawater was prepared by diluting seawater with tap water, while the 35 psu seawater was prepared by adding artificial sea salt to the natural seawater. The temperature was kept at 24 °C throughout the whole experiment. The clams were cultured in different salinity levels for 72 h. The hemolymph sample from five clams was collected at different sampling times (0, 2, 4, 8, 12, 24, 48, and 72 h) post-exposure. The osmotic pressure of hemolymph was measured by using the Gonotec OSMOMAT 3000 (Gonotec GmbH, Berlin, Germany). The gill samples from fifteen clams were collected simultaneously. Samples were rapidly frozen in liquid nitrogen and stored at −80 °C prior to the total RNA extraction and following qRT-PCR analysis. In addition, gill, digestive gland, and adductor muscle samples were collected separately from three groups at 48 h for the following FISH analysis.

### 2.4. RNA Isolation and Reverse Transcription

Total RNA was extracted from the clam samples collected at different tissues and developmental stages, using RNAiso Plus reagent (TaKaRa, Dalian, China). The RNA was then treated with RNase-free DNase (TaKaRa, Dalian, China) to eliminate contaminating genomic DNA. The quantity and quality of purified RNA were assessed by spectrophotometry and electrophoresis on 1% agarose gels. RNA samples with OD 260/280 of 2.0 and most intense smear between 400 bp and 4 kb were processed further for the first-strand complementary DNA (cDNA) synthesized. The cDNA was synthesized by using Prime Script™ Reverse Transcription Kit (TaKaRa, Dalian, China). The synthesized cDNA product was 10-times diluted and stored at −20 °C until further analysis.

### 2.5. Cloning of Sc-AQP1, Sc-AQP8, and Sc-AQP11 cDNA Sequence

To amplify the cDNA ends (5′ and 3′ RACE) of *Sc-AQP1*, *Sc-AQP8*, and *Sc-AQP11*, three pairs of primers were designed according to the known transcriptome data of *S. constricta* (SRX6707291) and listed in Table 1. The first-strand cDNA was synthesized by using the SMART RACE cDNA Amplification Kit (Clontech, Mountain View, CA, USA), and the touchdown PCR program was performed by following the manufacturer's instruction. The targeted PCR products were then purified and cloned into pEasy-T1 vector (Transgen, Beijing, China). Positive clones were sent to the company (Sangon, Shanghai, China) for sequencing.

### 2.6. Bioinformatics Sequence Analysis

The nucleotide sequences of *Sc-AQP1*, *Sc-AQP8*, and *Sc-AQP11* were analyzed by using the DNAStar 7.0 software. The protein sequence and the open reading frame (ORF) were predicted by using the ExPASy translation tools (http://www.expasy.ch/, accessed on 1 October 2020.) and ORF Finder (https://www.ncbi.nlm.nih.gov/orffinder/, accessed on 3 October 2020), respectively. The signal peptide and domain were predicted by using the Signal P software (http://www.cbs.dtu.dk/services/Signal%20P, accessed on 15 October 2020) and Simple Modular Architecture Research Tool (http://smart.embl-heidelberg.de/, accessed on 20 December 2020), respectively. The physicochemical characteristics, protein 3D structure, ligand-binding site, and transmembrane helices of the protein sequence were predicted according to the method mentioned in previous research [15].

**Table 1.** Primers used for *Sc-AQP1*, *Sc-AQP8*, and *Sc-AQP11* gene cloning; qRT-PCR; and RNA interference.

| Primer | Primer Sequence (5′-3′) | Purpose | Amplicon Size (bp) |
|---|---|---|---|
| 5′RACE-AQP1 | CCAGCAGTCCACCCACAATAGGC | 5RACE | 1126 |
| 5′RACE-AQP8 | CCAGGCTCCACAGTCGTGTCACCAAT | 5′RACE | 761 |
| 5′RACE-AQP11 | TCCCCCCACCAAGACTGCCGAAT | 5′RACE | 879 |
| 3′RACE-AQP1 | ACACCAGCAACTCCAGCACCCT | 3′RACE | 1167 |
| 3′RACE-AQP8 | TGGTGACACGACTGTGGAGCCTGG | 3′RACE | 1498 |
| 3′RACE-AQP11 | CGGCTTTTGGCTCTTTATCGCTGTT | 3′RACE | 1381 |
| RT-AQP1F | CACCAGCAACTCCAGCCA | qRT-PCR | 142 |
| RT-AQP1R | CAGGACCGCCCTCCATAA | qRT-PCR | |
| RT-AQP8F | CATCTGTCCCCGATTATTGGT | qRT-PCR | 90 |
| RT-AQP8R | GGTGAAGACGAGAACCAGTGT | qRT-PCR | |
| RT-AQP11F | TGCCTGAACCAATCAAAAC | qRT-PCR | 120 |
| RT-AQP11R | CAGCGATAAAGAGCCAAAA | qRT-PCR | |
| 18SF | TCGGTTCTATTGCGTTGGTTTT | qRT-PCR | 121 |
| 18SR | CAGTTGGCATCGTTTATGGTCA | qRT-PCR | |
| dsRNAi- AQP1F1 | TAATACGACTCACTATAGGGGTCACCCCTAGCCGTCTACA | RNAi | 320 |
| dsRNAi- AQP1R1 | TAATACGACTCACTATAGGGAGCAGTCCACCCACAATAGG | RNAi | |
| dsRNAi- AQP8F1 | TAATACGACTCACTATAGGGTCGGGGTGACATTGTTTGTA | RNAi | 316 |
| dsRNAi- AQP8R1 | TAATACGACTCACTATAGGGCCAATAATCGGGGACAGATG | RNAi | |
| dsRNAi- AQP11F1 | TAATACGACTCACTATAGGGTGATGCCTGAACCAATCAAA | RNAi | 379 |
| dsRNAi- AQP11R1 | TAATACGACTCACTATAGGGCCGTAAAAGAACGCCACATT | RNAi | |
| siRNAi- AQP1F1 | GATCACTAATACGACTCACTATAGGGTTCAGATGTTCGGACACATTTCATT | RNAi | 22 |
| siRNAi- AQP1R1 | AATGAAATGTGTCCGAACATCTGAAGTGATC | RNAi | |
| siRNAi- AQP1F2 | AATTCAGATGTTCGGACACATTTCAGTGATC | RNAi | 22 |
| siRNAi- AQP1R2 | GATCACTAATACGACTCACTATAGGGGCTCCGAAAAATGTTCTGTTGGCTT | RNAi | |
| siRNAi- AQP8F1 | GATCACTAATACGACTCACTATAGGGCTGCTGAATGAAGAACATCGAACTT | RNAi | 22 |
| siRNAi- AQP8R1 | GCTCCGAAAAATGTTCTGTTGGCGTGATC | RNAi | |
| siRNAi- AQP8F2 | AACTGCTGAATGAAGAACATCGAACGTGATC | RNAi | 22 |
| siRNAi- AQP8R2 | GATCACTAATACGACTCACTATAGGGTTCGATGTTCTTCATTCAGCAGTT | RNAi | |
| siRNAi- AQP11F1 | GATCACTAATACGACTCACTATAGGGCCAACAGAACATTTTTCGGAGCTT | RNAi | 22 |
| siRNAi- AQP11R1 | AAGCTCCGAAAAATGTTCTGTTGGCGTGATC | RNAi | |
| siRNAi- AQP11F2 | AAGCCAACAGAACATTTTTCGGAGCGTGATC | RNAi | 22 |
| siRNAi- AQP11R2 | GATCACTAATACGACTCACTATAGGGGCTCCGAAAAATGTTCTGTTGGCTT | RNAi | |

### 2.7. Multiple Sequences Alignment and Phylogenetic Analysis

The homology search of the amino acid sequence of *Sc-AQP1*, *Sc-AQP8*, and *Sc-AQP11* was performed by using BlastX algorithm at the National Center for Biotechnology Information (http://www.ncbi.nlm.nih.gov/BLAST/, accessed on 7 January 2021). A total of 80 *AQP* sequences were used to construct the phylogenetic tree, and the detailed information of these sequences was listed in Table S1. The multiple sequence alignments of *Sc-AQP*s were aligned by using ClustalW analysis program, and the aligned sequences were subsequently imported in MEGA 7.0 software to construct a phylogenic tree. Only 1000 data bootstrap replications calculated in the neighbor-joining method were used for the phylogenic tree construction.

### 2.8. qRT-PCR Analyses

To determine the relative expression levels of *Sc-AQP1*, *Sc-AQP8*, and *Sc-AQP11* genes, qRT-PCR was carried out by using a Light Cycler 480 instrument (Roche, Basel, Switzerland) with the synthesized total cDNA as a template. Primer sequences for each gene were designed by using the program of Primer 5 software and are listed in Table 1. Each qRT-PCR reaction was performed in a total volume of 20 μL mixture containing 0.8 μL of cDNA, 1 μL of forward primer, 1 μL of reverse primer, 7.2 μL of PCR-grade water, and 10 μL of 2× SYBR Green Supermix (Bio-Rad, Hercules, CA, USA). All qRT-PCR reactions were set up as follows: pre-incubation step at 95 °C for 5 min, followed by 40 cycles of denaturing at 95 °C for 10 s and annealing at 61 °C for 20 s. A melting curve analysis was conducted to confirm that a single PCR product was produced by each pair of primers during the reaction. All samples were measured in technical triplicate. The

relative expression level of each gene was normalized to 18S rRNA gene and subsequently calculated according to the standard $2^{-\Delta\Delta CT}$ method [16].

### 2.9. RNA Interference

Double-stranded RNAs (dsRNA) and small interfering RNA (siRNA) specific to the *Sc-AQP1*, *Sc-AQP8*, and *Sc-AQP11* were designed by using the SnapDragon-dsRNA Design tool and Small Interfering RNA (DSIR) webtool (https://www.flyrnai.org/cgi-bin/RNAi_find_primers.pl, accessed on 11 March 2021), respectively. The dsRNA and siRNA were generated by using the T7 RNAi Transcription Kit (Vazyme, Nanjing, China) with primers listed in Table 1. The synthesized dsRNA and siRNAs were subsequently purified by using Monarch RNA Cleanup Kit (NEB, Ipswich, MA, USA), diluted to 1 µg/µL, and stored at −80 °C prior to the injection. Two hundred and ten clams from normal salinity group (18 psu) were randomly divided into 7 groups, and 30 clams in each group were intramuscularly (adductor muscle) injected with different solutions as follows: 100 µL of dsRNA-*Sc-AQP1* and 100 µL of siRNA-*Sc-AQP1* for *Sc-AQP1*-silenced groups, 100 µL of dsRNA-*Sc-AQP8* and 100 µL of siRNA-*Sc-AQP8* for *Sc-AQP8*-silenced groups, 100 µL of dsRNA-*Sc-AQP11* and 100 µL of siRNA-*Sc-AQP1* for *Sc-AQP11*-silenced groups, and 100 µL of diethylpyrocarbonate (DEPC)-treated water for the control group. Five clams were collected from each group at every sampling time (0, 12, 24, 48, 72, and 96 h). The gill, digestive gland, and adductor muscle were sampled to detect the relative expression of *Sc-AQP1*, *Sc-AQP8*, and *Sc-AQP11*, respectively. The osmotic pressure of hemolymph was measured simultaneously by using the Gonotec OSMOMAT 3000 (Gonotec GmbH, Berlin, Germany).

### 2.10. Fluorescence In Situ Hybridization

RNA probes labeled with digoxigenin (DIG) were prepared from the cDNA of *Sc-AQP1*, *Sc-AQP8*, and *Sc-AQP11* by using a T7 High-Efficiency Transcription Kit (Trans-Gen, Beijing, China) and DIG RNA labeling kit (Roche, Basel, Switzerland). The FISH analysis of *Sc-AQP1*, *Sc-AQP8*, and *Sc-AQP11* was conducted on the paraffin section of gill, digestive gland, and adductor muscle, respectively. The preparation of the paraffin section and the subsequent procedure of FISH were handled according to the method mentioned in previous research [17]. The expression of each gene in different tissues was observed by using a fluorescence microscope (Nikon, 80i, Tokyo, Japan).

### 2.11. Statistical Analysis

All statistical analyses were performed with the SPSS 13.0 statistical package (IBM, Manassas, VA, USA). All experimental data were presented as mean ± standard error (SE). The Kolmogorov–Smirnov test was used to inspect the normality and homogeneity of variance of all the data. Data with normal distribution were statically analyzed by using one-way ANOVA with the Least Significance Difference (LSD) test. For all comparisons, a *p*-value < 0.05 was considered statistically significant.

## 3. Results

### 3.1. Sequence Features of Sc-AQP1, Sc-AQP8, and Sc-AQP11

The entire genome sequences of *Sc-AQP1*, *Sc-AQP8*, and *Sc-AQP11* were all obtained by using 3′RACE and 5′RACE amplification (Figure 1). The completed cDNA sequence of *Sc-AQP1*, *Sc-AQP8*, and *Sc-AQP11* was deposited in GenBank under accession numbers MN186579, MN186580, and MN186581, respectively. The sequence features of each gene are listed in Table 2. The complete ORF of each gene included a downstream in-frame stop codon and a polyadenylation signal (AATAA motif) before the poly-A tail. According to the topology prediction of these proteins, three *Sc*-AQPs contained six putative transmembrane domains, five connecting loops, and a cytoplasmic N- and C-terminal domain. In addition, *Sc-AQP1*, *Sc-AQP8*, and *Sc-AQP11* contained 6, 8, and 4 ATTTA motifs in their 3′-UTR, respectively. As displayed in Figure 2, typical AQP structures including monomers

containing water pores and tetramers were observed in the three *Sc*-AQPs. Notably, the two channel-forming NPA signature motifs were found at amino acid positions 95–97 and 210–212 in *Sc-AQP1* and positions 75–77 and 197–199 in *Sc-AQP8*. In contrast to *Sc-AQP1* and *Sc-AQP8*, the *Sc-AQP11* had only one highly conserved NPA sequence, with other NPA motifs replaced by Asn-Pro-Cys (NPC) motifs (Figure 3).

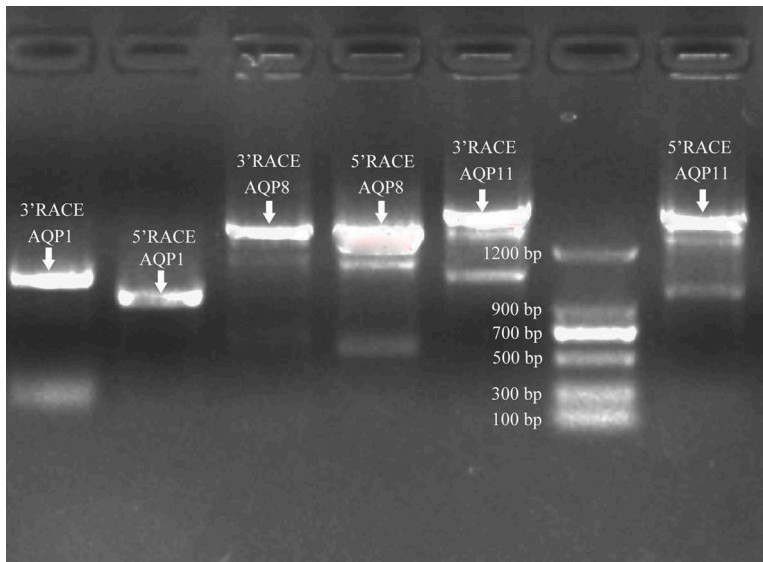

**Figure 1.** *Sc-AQP1*, *Sc-AQP8*, and *Sc-AQP11* 3′RACE and 5′RACE amplifications.

**Table 2.** Summary of sequence features of *Sc-AQP1*, *Sc-AQP8*, and *Sc-AQP11*.

| Sequence Features | *Sc-AQP1* | *Sc-AQP8* | *Sc-AQP11* |
|---|---|---|---|
| Gen Bank ID | MN186579 | MN186580 | MN186581 |
| cDNA length (bp) | 1546 | 2235 | 1713 |
| ORF (bp) | 900 | 771 | 843 |
| Length of amino acids (aa) | 299 | 256 | 280 |
| Molecular weight (kDa) | 32.34 | 26.80 | 31.52 |
| Theoretical pI | 6.19 | 6.37 | 5.89 |
| GRAVY | 0.464 | 0.738 | 0.415 |
| Asp + Glu | 21 | 17 | 25 |
| Arg + Lys | 19 | 15 | 20 |
| Instability index | 35.97 | 31.13 | 31.35 |
| Aliphatic index | 97.53 | 126.87 | 99.61 |
| 5′-UTR (bp) | 405 | 336 | 93 |
| 3′-UTR (bp) | 240 | 1128 | 777 |
| ATTTA motif | 6 | 8 | 4 |
| AATAA motif | 3 | 3 | 6 |
| NPA motifs | 2 | 2 | 1 |
| NPC motifs | 0 | 0 | 1 |
| MIP motifs | 1 | 1 | 1 |
| Transmembrane helix | 6 | 6 | 6 |
| loops | 5 | 5 | 5 |

The BlastX analysis revealed that the three Sc-AQPs shared high amino acid sequence identity and similarity with mollusk counterparts compared to other species (Table 3). The Sc-AQP1 shared the highest identity (47%) and similarity (66.2%) with *Aplysia californica* (J. G. Cooper, 1863) (Table 3). The amino acid sequence of *Sc*-AQP8 shared 46.1–54.7% identity and 60.3–67.6% similarity to the AQP8 of mollusk, particularly with the highest identity (54.7%) and similarity (67.6%) to *C. gigas*. Similarly, the amino acid sequence of *Sc*-AQP11 had more than 65% similarity to that of AQP11 molecules of mollusk, excluding *Biomphalaria glabrata* (Say, 1818) (63.6%).

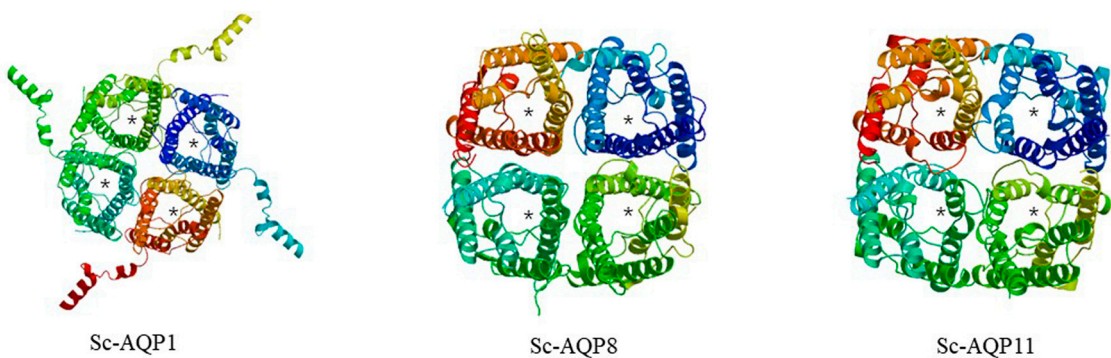

Sc-AQP1          Sc-AQP8          Sc-AQP11

**Figure 2.** Three-dimensional structure of a functional unit of AQP protein predicted by SWISS-MODEL. The functional unit of AQP is a tetramer with each monomer providing an independent channel pore that transports water across the cell membrane (∗).

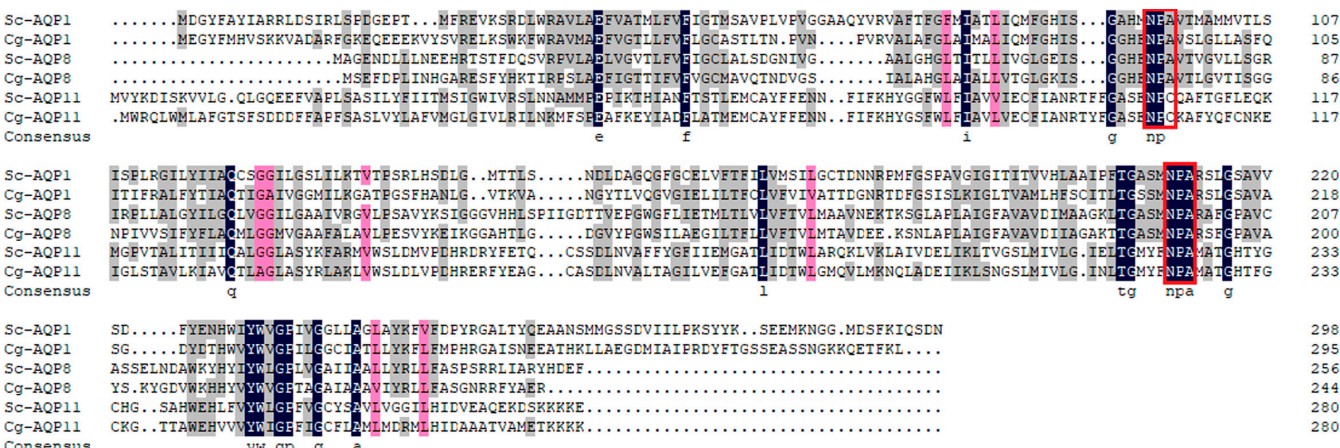

**Figure 3.** Alignment of the amino acid sequences among *Sc*-AQP1, Sc-AQP8, *Sc*-AQP11, *Cg*-AQP1, *Cg*-AQP8, and *Cg*-AQP11. The aligned sequences are as follows: Sc-AQP1, QIE08067.1; *Sc*-AQP8, QIE08068.1; *Sc*-AQP11, QIE08069.1; Cg-AQP1, XP_011446842.2; Cg-AQP8, XP_011436665.2; *Cg*-AQP11, XP_019923893.2. Cg indicated the *Crassostrea gigas* (Thunberg, 1793). Identical (dark blue) and similar (pink and gray) residues were indicated. Conserved NPA and NPC motifs are marked with the red frame.

The pairwise comparison showed that the amino acid sequence of *Sc*-AQP1 shared 27.5% identity and 39.6% similarity with that of *Sc*-AQP8, but that of *Sc*-AQP1 shared only 11.3% identity and 21.6% similarity with that of *Sc*-AQP11 (Table 3). Predicted protein chemistry properties also showed a similarity between *Sc*-AQP1 and *Sc*-AQP8 in molecular weight and pI. In addition, multiple sequence alignments revealed 10.4–46.1% identity and 20.2–65.5% similarity in the amino acid sequence of *Sc*-AQP1, *Sc*-AQP8, and *Sc*-AQP11 with *My*-AQP1, *My*-AQP8, and *My*-AQP11, respectively (Table 3).

## 3.2. Phylogenetic Tree Analysis

To elucidate the phylogenetic relationship of the three *Sc*-AQPs with other species, a phylogenetic tree was constructed. Results revealed that the phylogenetic tree could be divided into thirteen main clades (Figure 4). The phylogenic tree analysis of AQPs supported the idea that those newly identified *Sc*-AQPs can be classified into AQP1, AQP8, and AQP11.

**Table 3.** Amino acid identity and similarity of *Sc*-AQP1, *Sc*-AQP8, and *Sc*-AQP11 putative peptides as compared with other AQP1, AQP8, and AQP11 molecules.

| AQP * | Sc-AQP1 | | Sc-AQP8 | | Sc-AQP11 | |
|---|---|---|---|---|---|---|
| | Identity (%) | Similarity (%) | Identity (%) | Similarity (%) | Identity (%) | Similarity (%) |
| *Homo sapiens*—AQP1 | 35.9 | 49.3 | 30.6 | 43.5 | 12.3 | 22.5 |
| *Mus musculus*—AQP1 | 35.3 | 49.7 | 31 | 44.2 | 12 | 22.8 |
| *Gallus gallus*—AQP1 | 25.7 | 38.3 | 26.4 | 36.4 | 10.8 | 19 |
| *Xenopus tropicalis*—AQP1 | 34.6 | 49.7 | 29.8 | 42.8 | 10.3 | 22.1 |
| *Danio rerio*—AQP1 | 32.8 | 47.7 | 30.7 | 41.4 | 9.8 | 19 |
| Sc-AQP1 | 100 | 100 | 27.5 | 39.6 | 11.3 | 21.6 |
| *Aplysia californica*—AQP1 | 47 | 66.2 | 23.9 | 41.1 | 10.6 | 23.9 |
| *Mizuhopecten yessoensis*—AQP1 | 44.4 | 65.1 | 26.4 | 40.4 | 10.4 | 21.1 |
| *Helix pomatia*—AQP1 | 46.4 | 63.5 | 24.5 | 41.5 | 11.9 | 24.7 |
| *Crassostrea hongkongensis*—AQP1 | 43.7 | 59.6 | 27.8 | 39.8 | 11.2 | 19.9 |
| *Homo sapiens*—AQP8 | 22.8 | 37 | 37 | 54.3 | 13.3 | 23.5 |
| *Mus musculus*—AQP8 | 22.9 | 38.1 | 38.2 | 57.1 | 13.3 | 23.5 |
| *Gallus gallus*—AQP8 | 21.4 | 34.4 | 37.4 | 54 | 13.7 | 26.7 |
| *Alligator sinensis*—AQP8 | 21.2 | 35.4 | 37.4 | 53 | 12.3 | 22.3 |
| *Danio rerio*—AQP8 | 19.9 | 32.8 | 33.2 | 50.6 | 14.4 | 25 |
| Sc-AQP8 | 27.5 | 39.6 | 100 | 100 | 13.2 | 23.2 |
| *Crassostrea gigas*—AQP8 | 23.5 | 37.8 | 54.7 | 67.6 | 12.5 | 23.1 |
| *Mizuhopecten yessoensis*—AQP8 | 27.7 | 42.3 | 46.1 | 60.3 | 11.5 | 20.2 |
| *Pomacea canaliculata*—AQP8 | 26 | 38.9 | 50.2 | 64.1 | 13.3 | 21.4 |
| *Aplysia californica*—AQP8 | 28.5 | 39.8 | 51 | 64.1 | 12.2 | 22.7 |
| *Homo sapiens*—AQP11 | 13.5 | 24.5 | 14.1 | 24.1 | 18 | 31.7 |
| *Mus musculus*—AQP11 | 13.8 | 26 | 14.4 | 25.1 | 18 | 32.7 |
| *Gallus gallus*—AQP11 | 12 | 23.7 | 16 | 30.1 | 15.5 | 28.3 |
| *Xenopus laevis*—AQP11 | 12 | 21.5 | 14.4 | 29.1 | 19.7 | 30 |
| *Danio rerio*—AQP11 | 11.4 | 20.9 | 17.2 | 30.5 | 17.7 | 28.2 |
| Sc-AQP11 | 11.3 | 21.6 | 13.2 | 23.2 | 100 | 100 |
| *Crassostrea virginica*—AQP11 | 11.9 | 23.5 | 11.6 | 22.5 | 57.7 | 72.2 |
| *Crassostrea gigas*—AQP11 | 11.4 | 22 | 11.1 | 21.3 | 54.9 | 68.9 |
| *Biomphalaria glabrata*—AQP11 | 9.4 | 21.1 | 12.1 | 24.3 | 49.1 | 63.6 |
| *Mizuhopecten yessoensis*—AQP11 | 11 | 22.8 | 15.1 | 24.4 | 51 | 65.5 |

* The accession number of the AQP sequences used for this analysis are listed in Supplementary Table S1.

### 3.3. Spatiotemporal Expression Analysis of Sc-AQP1, Sc-AQP8, and Sc-AQP11

We quantified the mRNA expression levels of *Sc-AQP1*, *Sc-AQP8*, and *Sc-AQP11* in eight different tissues, including gill, intestine, kidney, digestive gland, siphon, mantle, foot, and adductor muscle. Results showed that the expressions of three *Sc-AQPs* mRNAs were detected in all examined tissues (Figure 5). *Sc-AQP1* was significantly expressed in the gill and intestine ($p < 0.01$) (Figure 5A). In contrast, the *Sc-AQP8* was significantly expressed in the digestive gland and gill, whereas the *Sc-AQP11* was significantly expressed in adductor muscle ($p < 0.01$) (Figure 5C,E).

The time-course expression revealed that the relative expression levels of *Sc-AQP1*, *Sc-AQP8*, and *Sc-AQP11* were expressed in all ten developmental stages (Figure 5B,D,F). The expression level of *Sc-AQP1* peaked up in the D-shaped larvae stage, followed by a downward trend, while that of *Sc-AQP8* was upregulated in the trochophore stage and peaked up in the umbo larvae stage, which was significantly higher than those detected in other stages ($p < 0.01$) (Figure 5B,D). A relatively higher expression of *Sc-AQP11* was detected in trochophore, D-shaped larvae, umbo larvae, and eyebot larvae stage (Figure 5F).

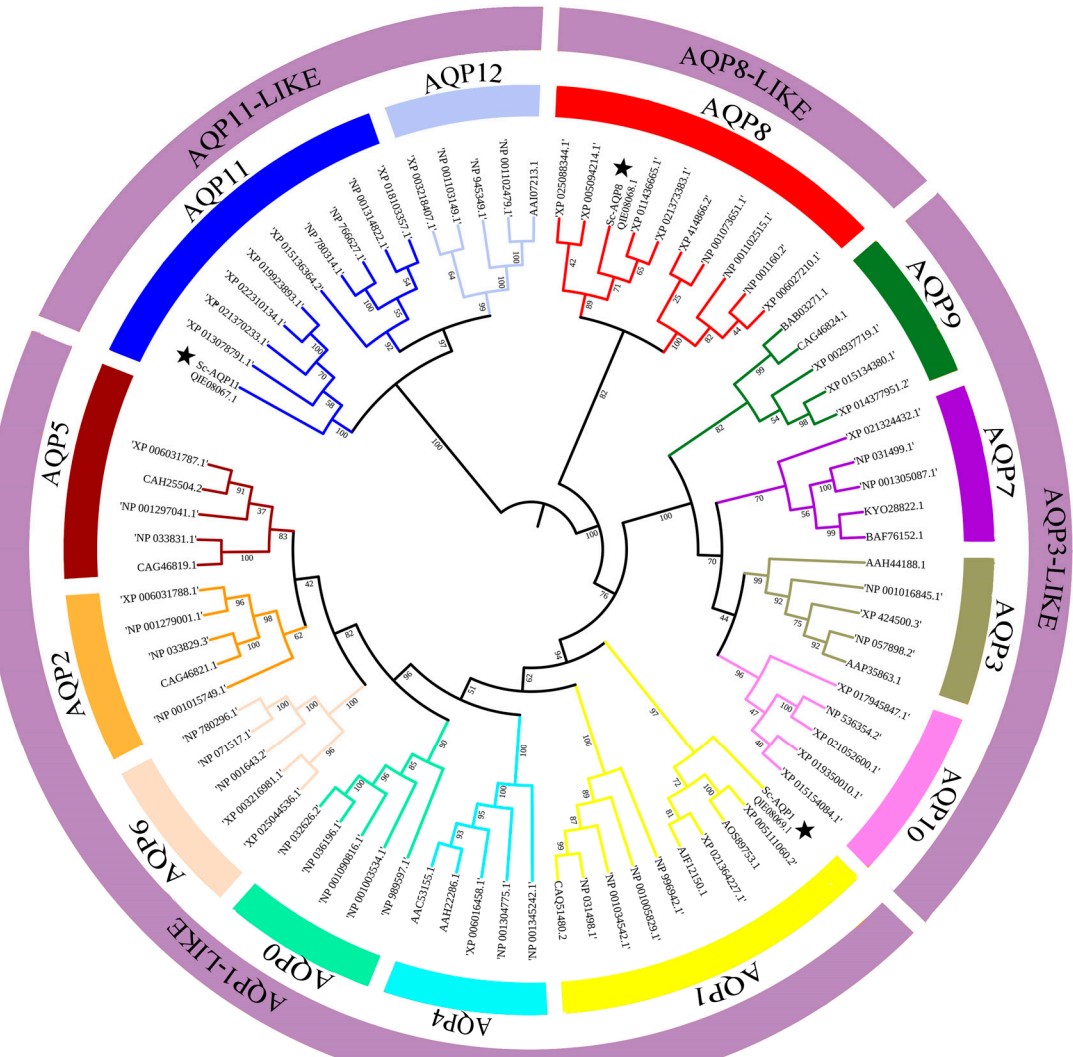

**Figure 4.** Phylogenetic analysis of three *Sc*-AQPs (*Sc*-AQP1, *Sc*-AQP8, and *Sc*-AQP11) with AQP family members of other species. The phylogeny tree was built by using amino acid sequences of 80 available AQPs. The AQP genes of *S. constricta* are labeled with star. The sequences are described by the GenBank accession number, and the detailed information of these sequences are included in Supplementary Table S1.

### 3.4. Quantitative Expression Analysis of Sc-AQP1, Sc-AQP8, and Sc-AQP11 mRNA after Salinity Challenge

To study the transcriptional response of *Sc-AQP* to salinity stress, razor clams were subjected to low-salt stress (3.5 psu) and high-salt stress (18 psu). Under low-salt and high-salt treatment, the *Sc-AQP1* expressions were induced and increased to 12.45-folds and 7.93-folds after 24 h of treatment, respectively, which were significantly higher than that of the control group ($p < 0.05$) (Figure 6A). For *Sc-AQP8*, its expression increased significantly at 24 h and reached the peak at 48 h (12.36-folds) under low-salt treatment ($p < 0.01$) (Figure 6B). However, it remained unchanged under high-salt treatment. The *Sc-AQP11* expression increased by 27.44- and 17.09-folds after 48 h of low-salt and high-salt treatment, respectively, which were significantly higher than that of the control group ($p < 0.01$) (Figure 6C). After 72 h, the RNA levels of *Sc-AQP* were all back to their initial levels.

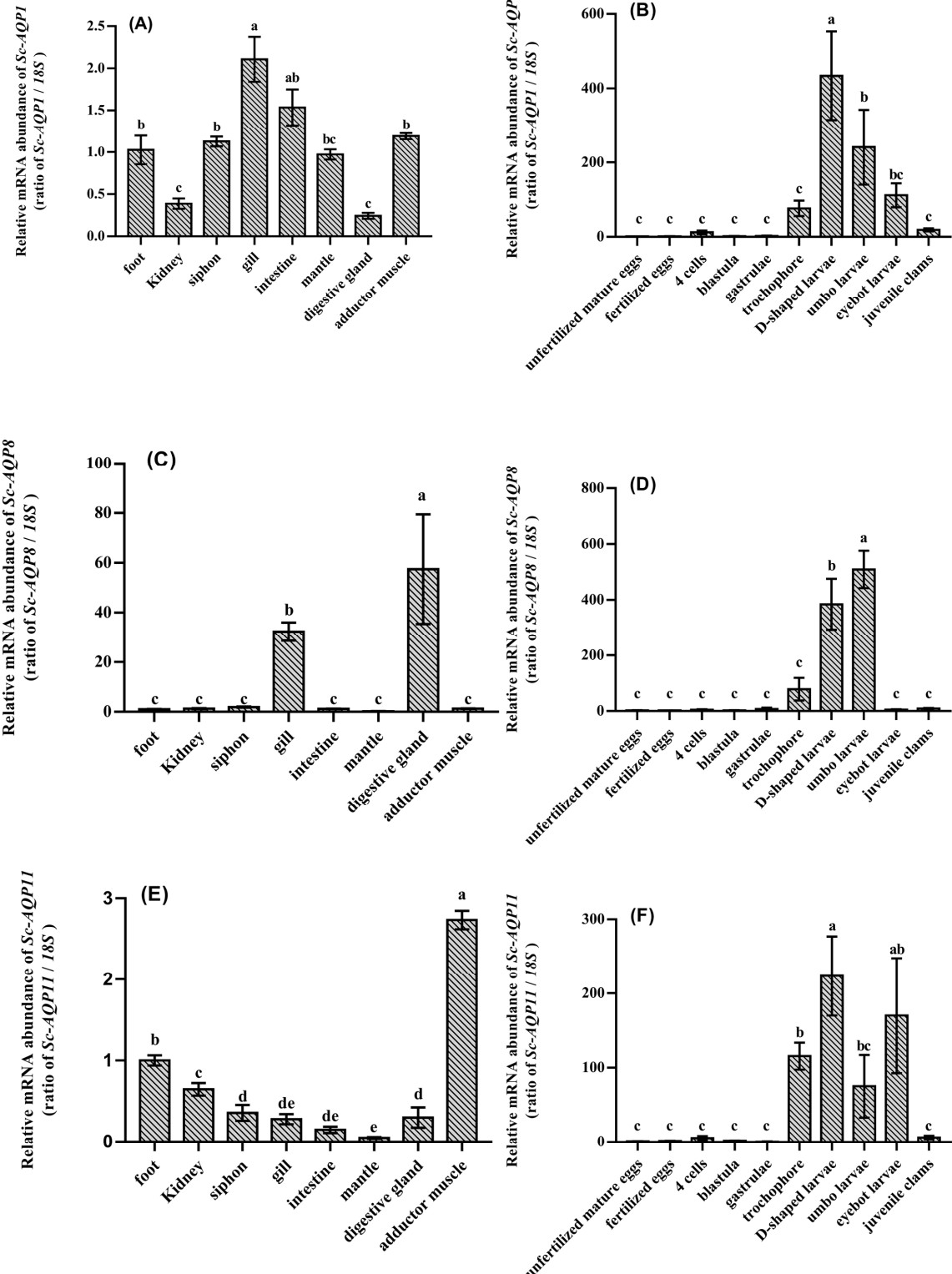

**Figure 5.** Quantitative expression analysis of *Sc-AQP1*, *Sc-AQP8*, and *Sc-AQP11* in different tissues (**A**,**C**,**E**) and developmental stages (**B**,**D**,**F**). Vertical bars represent the mean ± SE (*n* = 5) and mean ± SE (*n* = 200) in tissues and developmental stages analysis, respectively. SE represents independent biological replicates. Data were statistically analyzed by using one-way ANOVA, followed by the LSD test. Bars with different letters are significantly different (*p* < 0.01).

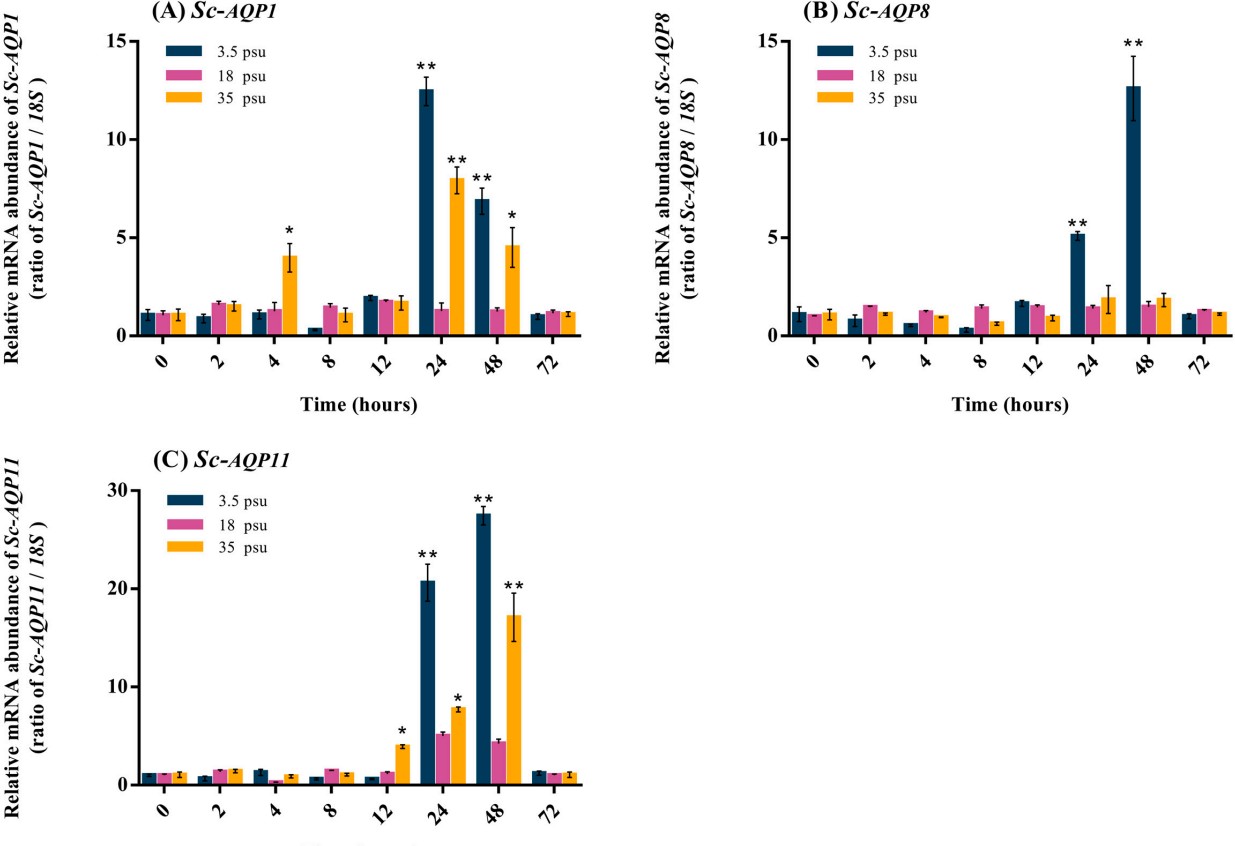

**Figure 6.** Temporal expressions of the *Sc-AQP1* (**A**), *Sc-AQP8* (**B**), and *Sc-AQP11* (**C**) after salinity challenge of 3.5, 18, and 35 psu. Vertical bars represent the mean ± SE (*n* = 15). Data were statistically analyzed by using one-way ANOVA followed by the LSD test. The asterisks (*) show a significant difference between the salinity challenge group (3.5 and 35 psu) and the control group (18 psu) at each sampling time. Note: * $p < 0.05$; ** $p < 0.01$.

### 3.5. Detection of the Osmotic Pressure of Hemolymph after Salinity Challenge

The osmotic pressure of hemolymph was detected in each sampling time post-exposure to the low- and high-salinity stimulation. The osmotic pressure in hemolymph significantly increased in the high-salinity group and peaked up to 1138.2 mO·smkg$^{-1}$ at 24 h ($p < 0.05$) (Figure 7). Conversely, the osmotic pressure significantly decreased within 96 h in the low-salinity group as compared with the control group ($p < 0.05$) (Figure 7).

### 3.6. Effects of dsRNA and siRNA on Sc-AQP1, Sc-AQP8, and Sc-AQP11 Expression and Osmotic Pressure

A qRT-PCR was performed to determine the expression levels of *Sc-AQP1*, *Sc-AQP8*, and *Sc-AQP11* in different tissues post RNA interference (Figure 8). The results revealed that dsRNA and siRNA significantly reduced the transcript level of *Sc-AQP1*, *Sc-AQP8*, and *Sc-AQP11*. The relative expression levels of the *Sc-AQP1* gene in gills were significantly decreased to the lowest level at 48 and 24 h post-injection with dsRNA-*Sc-AQP1* and siRNA-*Sc-AQP1*, respectively (Figure 8A). The relative expression levels *Sc-AQP8* gene in digestive gland reached the lowest level at 24 h post-injection with dsRNA-*Sc-AQP8* and siRNA-*Sc-AQP8*, respectively (Figure 8B). Post-injection with dsRNA-*Sc-AQP11* and siRNA-*Sc-AQP11*, the *Sc-AQP11* gene expression in adductor muscle significantly decreased to 12% and 28% of its initial level, respectively ($p < 0.01$) (Figure 8C).

The osmotic pressure in hemolymph first decreased from 24 and 12 h and then reached the lowest values of 563 and 543 mO·smkg$^{-1}$ at 96 h after interference with dsRNA-AQP1 and siRNA-AQP1, respectively (Figure 8D). In contrast, the osmotic pressure in hemolymph began

to decrease from 48 and 24 h and then reached the lowest values of 599 and 583 mO·smkg$^{-1}$ at 96 h after interference with dsRNA-AQP8 and siRNA-AQP8, respectively (Figure 8D). Interestingly, no change was observed in osmotic pressure in hemolymph after interference with dsRNA-AQP11 and siRNA-AQP11 (Figure 8D).

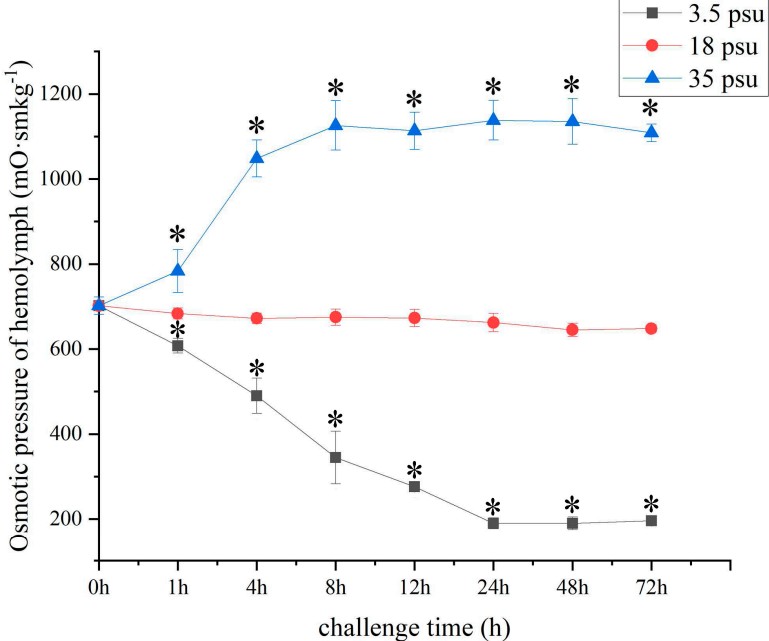

**Figure 7.** Osmotic pressure changes in *S. constricta* post-exposure to the low- and high-salinity environment. Vertical bars represent the mean ± SE (*n* = 5). Data were statistically analyzed by using one-way ANOVA, followed by the LSD test. The asterisks (*) show a significant difference between the experimental group (low- and high-salinity group) and the control group at each sampling time. Note: * $p < 0.05$.

### 3.7. Fluorescence In Situ Hybridization of Sc-AQP1, Sc-AQP8, and Sc-AQP11

We further performed the FISH analysis to confirm the expression level of *Sc-AQP1*, *Sc-AQP8*, and *Sc-AQP11* in different tissues in response to the low- and high-salinity pressure. A stronger signal of *Sc-AQP1* was detected at the inner side of gill filaments post low-salt and high-salt treatment (Figure 9). Similarly, low-salt and high-salt stress also induced a stronger signal of *Sc-AQP11* at the surface of the adductor muscle (Figure 9). For *Sc-AQP8*, the stronger signal was only observed at the digestive gland post low-salt treatment (Figure 9).

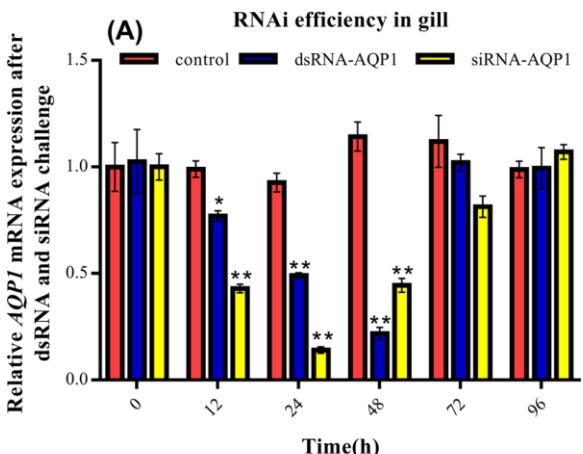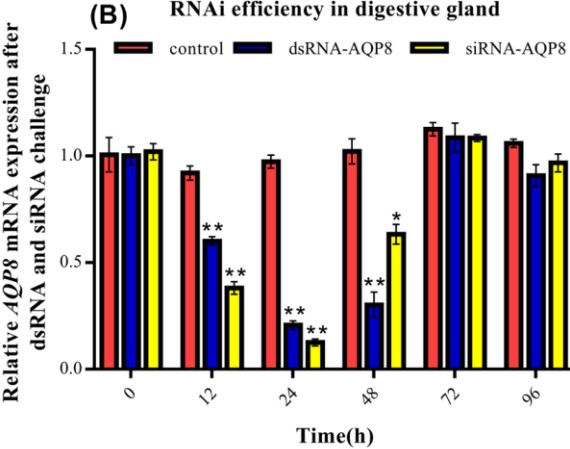

**Figure 8.** *Cont.*

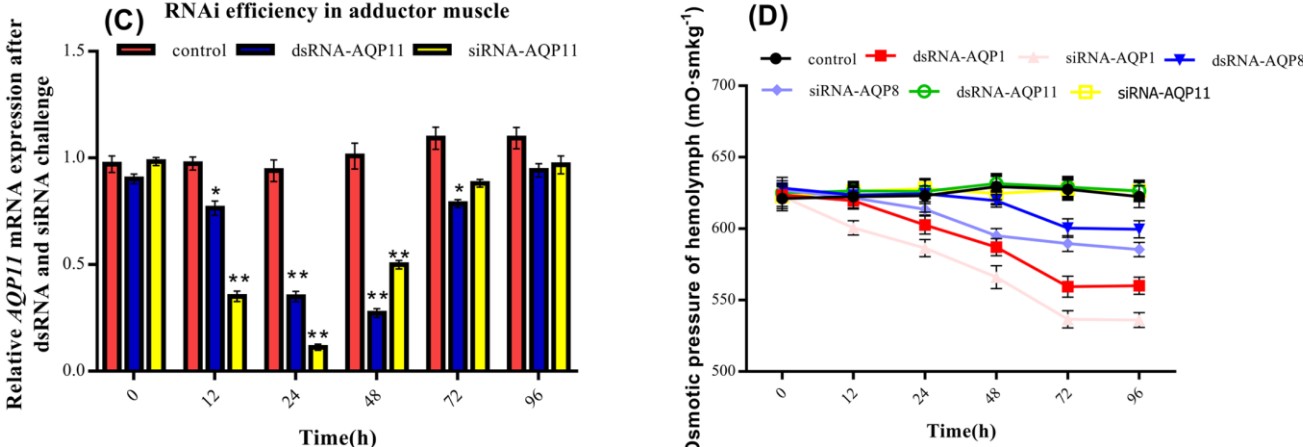

**Figure 8.** Relative expression levels of *Sc-AQP1* (**A**), *Sc-AQP8* (**B**), and *Sc-AQP11* (**C**) and osmotic pressure changes (**D**) in *S. constricta* post-injection with DEPC-treated water and corresponding dsRNA and siRNA. Vertical bars represent the mean ± SE (*n* = 5). Data were statistically analyzed by using one-way ANOVA, followed by the LSD test. The asterisks (*) show a significant difference between the RNA interference group (dsRNA and siRNA) and the control group at each sampling time. Note: * $p < 0.05$; ** $p < 0.01$.

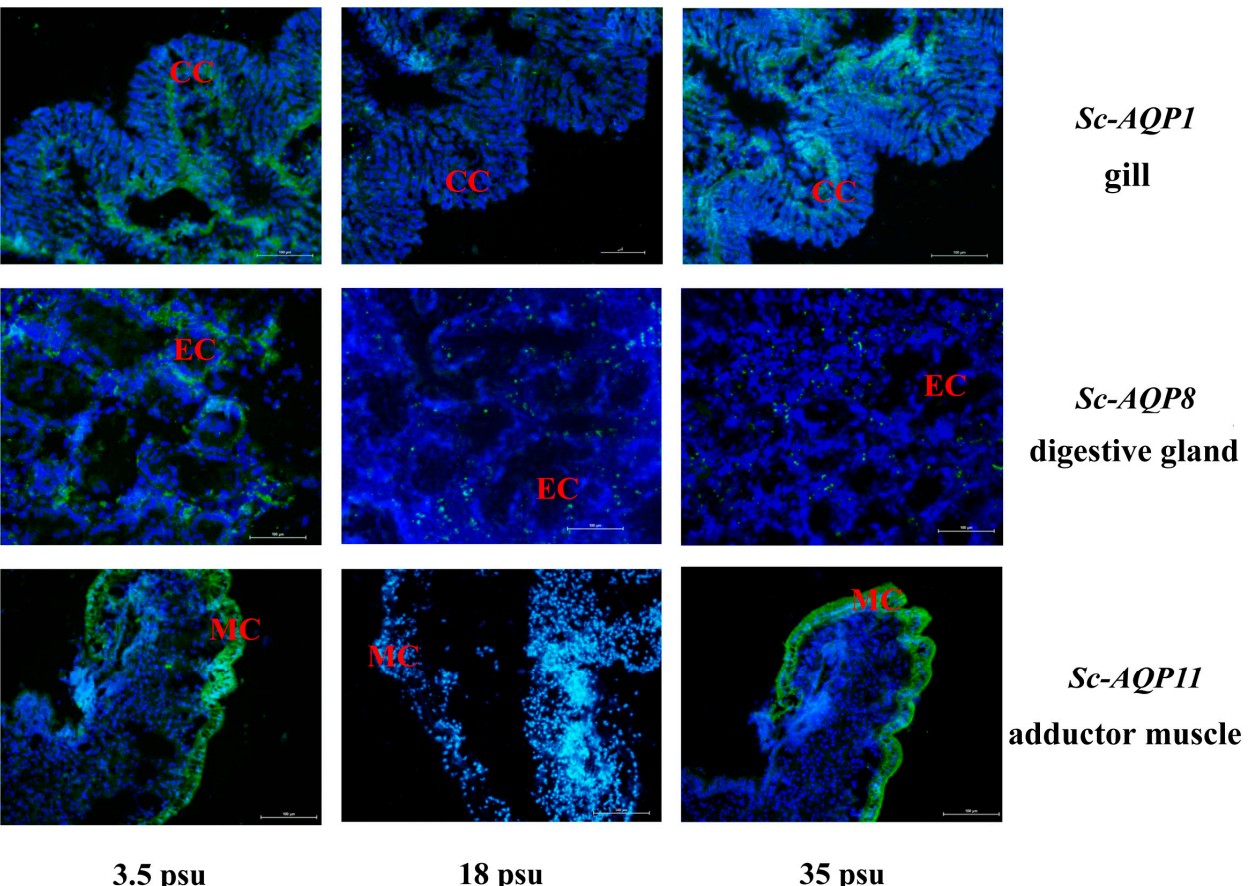

**Figure 9.** Immunofluorescence of *Sc-AQP1*, *Sc-AQP8*, and *Sc-AQP11* in different tissues. The positive signal of *Sc-AQPs* genes is indicated in green, and nuclei were stained with DAPI in blue. CCs, columnar cells. ECs, endothelial cells. MCs, muscle cells. Scale bars are 100 μm.

## 4. Discussion

It is well-known that bivalves tolerate a wide range of environmental stress through adaptation mechanisms. Thus, *S. constricta* is able to regulate their extracellular and intracellular fluids under salinity changes. Notably, osmotic pressure regulation is a complex biological process, in which many genes belonging to the *AQP* family and also many other types of genes, e.g., ion channels, are involved [18]. AQP is a specialized channel that rapidly transports water and other small solutes within different organs [19]. The channels facilitate the rapid movement of water across cell membranes [20]. The permeability of some cells to water can change rapidly due to the fact that some AQP properties can be activated/inactivated by phosphorylation, or translocated in and outside the cell membrane [21]. So far, at least 30 *AQP* genes were discovered in other mollusks, including *C. gigas*, *C. hongkongensis* (Lam et Mortan, 2003), and *C. virginica* (Gmelin, 1791), while no *AQP* genes were identified from *S. constricta*. In this study, we cloned and characterized three cDNAs encoding *AQPs* from the gill of *S. constricta*.

The AQP family commonly contained five loops, termed as loops A–E, and are responsible for connecting membrane-spanning α-helices [22]. Within them, two loops, namely the cytosolic loop (loop B) and the extra-cytosolic loop (loop E), contain conserved Asn-Pro-Ala (NPA) motifs [22]. The interaction between these conserved NPA motifs and the adjacent residues located on loop B and loop E is vitally important for water transport across the membrane [23]. Notably, some AQPs show the N-terminal NPA modified into NPC to form tetramers [24]. Consistent with the notion, the *Sc*-AQP1 and *Sc*-AQP8 detected here contained two conserved NPA motifs, which caped the end of the two half helices and lay at the middle of the permeation channel. Thus, we inferred that *Sc*-AQP1 and *Sc*-AQP8 can form a constriction that appears to act mainly as size exclusion filters. Notably, *Sc*-AQP11 had a unique amino acid sequence that included an NPC motif which corresponded to the N-terminal NPA signature motif of conventional AQPs. The substitution of cysteine residues with alanine residues in AQP11 results in the suppression of water transport efficiency [24]. Nevertheless, previous research revealed that a key amino acid residue (Tyr83) facing the channel pore might be responsible for its slow but constant water permeable function of AP11 [25].

The three *Sc*-AQPs had higher identity and similarity with mollusks than mammalian counterparts, suggesting their close evolutionary relationship. It has been proposed that AQP11 and AQP12 are the most distantly related paralogs and share only 20% homology with AQP family members [26]. A similar finding was observed in *Sc*-AQP11 in this study. Comparisons of AQP sequences from *C. gigas* have revealed that some domains within *Sc*-AQPs remained unchanged through evolution, suggesting that a similar molecular mechanism of water transport among the species exists [27]. For most kinds of mollusks, such as snails, *Pinctada fucata martensii* (Dunker, 1872), and *C. hongkongensis*, the AQPs have been proved to be ubiquitous proteins and play important roles in many essential water transport functions [28,29]. Similarly, the phylogenetic tree also showed that there existed a strictly evolutionary relationship in *Sc*-AQPs with arthropods and fishes [30,31]. For the crustaceans, in addition to the osmotic pressure regulating, AQPs expression also changes during the moult cycle of a decapod crustacean, together with the regulation of cell volume with the participation of AQPs [32].

In order to further confirm their roles in water transport, multiple experiments, including tissue distribution analysis, salinity stress analysis, and RNA interference, were conducted. The vital role of AQP1 contributed to water transport has been verified in many researches [33,34]. For example, Pallone et al. (2000) proved that deletion of *AQP1* leads to a urinary concentrating deficiency in outer medullary descending vasa recta (OMDVR) water transport [33]. Deane et al. (2011) also claimed that AQP1a played a key role in releasing water through the basolateral membrane of the cells in order to avoid cell swelling when confronted with hypo-osmotic stimulation [34]. In this study, *Sc-AQP1* showed a wide tissue distribution similar to those observed in other marine species [31,35,36]. In addition, *Sc-AQP1* seemed to be sensitive to the salinity adaptation, because its transcript

expression was significantly upregulated post hyposaline and hypersaline acclimation. FISH also indicated that its expression in gill was higher than in the control group after changing water salinity. Knockdown of the *Sc-AQP1* directly reduced the osmotic pressure of hemolymphoid, which was similar to previous research [37]. RNAi gene silencing of *CLAQP1* in *Cimex lectularius* (Linnaeus, 1758) significantly reduced water excretion [37]. Thus, we believed *Sc-AQP1* in the gill was important for *S. constricta* to transport water and regulate osmotic pressure.

Similarly, the Sc-AQP8 gene was also detected in all test organs, especially in the digestive gland and gill. The organs belonging to the digestive gland are highly involved in water transport, and AQP8 is important in this process [38,39]. For example, Laforenza et al. (2005) indicated that AQP8 plays a major role in water movement through the colon [38]. Ferri et al. (2003) also demonstrated its potential role in canalicular water secretion [39]. Moreover, Ferri et al. (2003) suggested an intracellular involvement of AQP8 in preserving cytoplasmic osmolality during glycogen metabolism and in maintaining mitochondrial volume [39]. The FISH analysis also showed that low-salinity challenge resulted in the upregulation of the *Sc-AQP8* gene in the digestive gland. Unfortunately, no research has been conducted to explain why *AQP8* transcript is highly expressed in the gill. We speculated that its high expression might be relative to the water transport and gas exchange in gill. Furthermore, our results showed that the *Sc-AQP8* in gill was sensitive to low salinity, implying its potential role in water transport. In addition, knockdown of *Sc-AQP8* led to the significant downregulation of *Sc-AQP8* expression, and the osmotic pressure of hemolymph decreased simultaneously. Thus, it was undisputed that the *Sc-AQP8* was highly related to water transport and osmotic pressure regulation. In the case of AQP8, apart from being water channel, it also supports the significant fluxes of $NH_3$ and $NH_4^+$ transport [40].

Furthermore, AQP11 owns a ubiquitous tissue distribution in rats and *Oryzias latipes* (Temminck and Schlegel, 1846), whereas it was found only in the gastrointestinal tract of zebrafish [41–43]. In this study, *Sc-AQP11* was found to be highly expressed in the adductor muscle of *S. constricta*, which was an important organ responsible for the ion transport from cells to the myostracum [44]. Thus, we believe this newly identified AQP11 is important for the ion and water transport, although there exists controversy over the functional studies of AQP11 in different tissues. Gorelick was unable to demonstrate the transport of water, glycerol, urea, or ions by AQP11 in *Xenopus oocyte* [45]. In contrast, Morishita found that AQP11 is essential for the proximal tubular function in the mice, which was correlated to the water transport [46]. In addition, the high expression of *Sc-AQP11* in gill in hyposaline and hypersaline acclimation further suggested its prominent role in water absorption.

Despite the lack of a complete osmotic adjustment mechanism, a majority of marine organisms initiate a set of adaptive mechanisms to deal with external stress, such as salinity changes [47]. Interestingly, we found the expression of transcript response to the hyposaline and hypersaline was distinct among *Sc-AQP1*, *Sc-AQP8*, and *Sc-AQP11*. The hypersaline acclimation resulted in an elevated expression of transcript of *Sc-AQP1* and higher osmotic pressure in hemolymph post-exposure for 24 h, and the suppression of *Sc-AQP1* mRNA expression led to the downregulation of osmotic pressure from 24 to 72 h. We speculated that the *Sc-AQP1* might play a role as outlet point for water drainage through the basolateral membrane, as has been verified in euryhaline silver sea bream [34]. Curiously, the hyposaline acclimation also resulted in an elevated expression of transcript of *Sc-AQP1*, which might be caused by osmotic pressure imbalance in gill. As the gas exchange organ of shellfish, gills need to balance the osmotic gradient between blood and surrounding water. Therefore, the gill tends to be highly permeable, meaning that the gill epithelial cells may require a more developed osmotic-pressure-regulation ability [48]. These findings suggest that the *Sc-AQP1* may be synthesized to excrete water entering the gill at high osmotic salinity, while its function appears to be limited at low salinity. In contrast, our results also showed that acclimation to salinities ranging from 18 to 35 psu did not significantly affect gill *Sc-AQP8* expression, and the *Sc-AQP8* was more sensitive

to low salinity than high salinity. To decrease the osmotic difference between the internal and external environment during salinity changes, chloride cells commonly change the structure for water flow and increase the flux of water molecules, and this was relative to the upregulation of *AQP8* expression [49]. We therefore propose that knockdown of the *Sc-AQP8* leads to the dysfunction of chloride cells and imbalance of osmotic pressure in the hemolymph. Similarly, the hyposaline and hypersaline acclimation resulted in an elevated expression of transcript of *Sc-AQP11*. Interestingly, although RNA interference resulted in the downregulation of *Sc-AQP11* expression in adductor muscle, it did not change the osmotic pressure in the hemolymph. This is because Sc-AQP11 may be involved in the slow but continuous movement of water across the membrane [25]. Intertidal organisms, including plants, fish, shellfish, and crabs, commonly form a species-dependent ontogenetic pattern to accommodate the fluctuation of environmental salinity [50]. Within these organisms, shellfish own a high osmoregulatory capacity to adapt to the estuarine conditions, especially in the larvae stage [51]. Nowadays, an increasing body of evidence suggests that AQP3, AQP7, AQP8, and AQP11 are the most abundant AQPs in sperm and are strongly activated in response to variations of osmolality [51,52]. Recent works have uncovered that AQP1b can facilitate water permeation and resultant swelling of the oocyte [53]. Similarly, *Sc-AQP1*, *Sc-AQP8*, and *Sc-AQP11* in this study were expressed in all developmental stages, suggesting that even larvae at earlier zoeal stages do have the osmoregulatory capacity. Moreover, our results demonstrated that the expression of *Sc-AQP1*, *Sc-AQP8*, and *Sc-AQP11* was highest at D-shaped larvae and umbo larvae stages. The possible explanation for this pattern is that *Sc-AQP1*, *Sc-AQP8*, and *Sc-AQP11* genes are needed to regulate the osmotic pressure when the clam larvae grow rapidly. On the contrary, the expression levels of *Sc-AQP1*, *Sc-AQP8*, and *Sc-AQP11* decreased gradually at the juvenile stage, but were significantly higher than that in the pre-developmental period. The reasons for difference in expression level of three *Sc-AQP* at different stages may be that the tissues and organs of the clam are formed at this time, but the growth and development are still relatively vigorous.

## 5. Conclusions

In summary, this study identified three new *AQPs* in response to salinity stress in *S. constricta*. The mRNA transcripts of the three AQPs displayed constitutively expressed in all examined tissues and developmental stages. Moreover, *Sc-AQP1*, *Sc-AQP8*, and *Sc-AQP11* displayed increased sensitivity to salinity change, with remarkable increasing expression in the gill post the exposure to the low salinity and high salinity, as evidenced by RNA interference and FISH analysis. These findings indicated that *Sc-AQP1*, *Sc-AQP8*, and *Sc-AQP11* were responsible for the osmoregulation. Collectively, our findings contributed to clarifying the role of *Sc-AQP* in salinity tolerance and provided foundational knowledge on the adaptive mechanism of razor clams under salinity stress.

**Supplementary Materials:** The following supporting information can be downloaded at https://www.mdpi.com/article/10.3390/fishes7020069/s1. Table S1: Detailed information of sequences used in phylogenetic tree construction.

**Author Contributions:** L.H. and Z.L. conceived of the project; L.H. and Y.D. designed the experiments; W.R. performed the experiments and data analysis; L.H. and W.R. wrote and revised the manuscript. All authors have read and agreed to the published version of the manuscript.

**Funding:** This work was supported by the National Key Research and Development Program of China (2020YFD0900802, 2018YFD0901405), National Natural Science Foundation of China (No. 31802322), and China Agriculture Research System of MOF and MARA.

**Institutional Review Board Statement:** In the present study, *Sinonovacula constricta* clams were collected and used. All experimental procedures were conducted according to the guidelines of the appropriate Institutional Animal Care and Use Committee (IACUC) of Zhejiang Wanli University, China (Approval Code: 20220124001).

**Data Availability Statement:** Supporting data are stored in Bankit of the Genbank database; the are relative ID numbers are *Sc-AQP1*, MN186579; *Sc-AQP8*, MN186580; and *Sc-AQP11*, MN186581. Other data presented in this study are available upon request from the corresponding author.

**Acknowledgments:** We would like to thank Ningbo Ocean and Fishery Science and Technology Innovation Base for the provision of experimental field.

**Conflicts of Interest:** The authors declare no conflict of interest.

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
