# Peer review of "Molecular Characterization of Aquaporins Genes from the Razor Clam Sinonovacula constricta and Their Potential Role in Salinity Tolerance"

_fishes, doi:10.3390/fishes7020069_

Round 1

Reviewer 1 Report

General comments:

The evaluated manuscript brings results of an interesting study conducted on molecular characterization of aquaporins genes from the razor clam Sinonovacula constricta and their potential role in salinity tolerance. Although several studies have been conducted on effects of high salinity stress on the survival, gill tissue, enzyme activity and free amino acid content, such studies are always welcome. The work is well structured. There are only minor formatting errors and/or typos in the text. Based on the small observations below, I can recommend publication of the study in Fishes after minor revision.

Specific comments:

The abstract should be a total of about 200 words maximum. The authors’ is over 300 words. Please summarize the information.

Lines 80-93. In total, how many bivalves were used? How were main chemical and physical parameters measured? Please specify.

Author Response

Dear Reviewer:

Thanks for your recognition of our paper. We have considered carefully the valuable suggestions from you and tried our best to revise our manuscript. We hope it will meet with your approval. Revisions resulted from your comments are visible in the paper using the “Track Changes” function in Microsoft Word. The detailed responses are listed as attachments. 

Reviewer 2 Report

The study entitled “Molecular characterization of aquaporins genes from the razor clam Sinonovacula constricta and their potential role in salinity tolerance” by Ruan et al. is a detailed and focused study on the osmotic physiology of this bivalve. The authors have done a great job integrating levels of all biological responses in their manuscript. I believe that this research adds a significant part in this field of science and therefore I suggest its publications after the comments which are found on the attached pdf are addressed. Moreover, the authors should correct some grammatical and syntactical errors scattered in the text. I have done so myself, but the text needs a more thorough grammatical editing.

Author Response

(The authors gave the same response as above.)
